# Replacement of Valproic Acid with New Anti-Seizure Medications in Idiopathic Generalized Epilepsy

**DOI:** 10.3390/jcm11154582

**Published:** 2022-08-05

**Authors:** Ayataka Fujimoto, Hideo Enoki, Keisuke Hatano, Keishiro Sato, Tohru Okanishi

**Affiliations:** 1Department of Neurosurgery, Seirei Hamamatsu General Hospital, 2-12-12 Sumiyoshi, Nakaku, Hamamatsu 430-8558, Japan; 2Comprehensive Epilepsy Center, Seirei Hamamatsu General Hospital, 2-12-12 Sumiyoshi, Nakaku, Hamamatsu 430-8558, Japan; 3Division of Child Neurology, Department of Brain and Neurosciences, Faculty of Medicine, Tottori University, Yonago 683-8503, Japan

**Keywords:** seizure frequency, myoclonus, anti-seizure medication (ASM), idiopathic generalized epilepsy (IGE), replacement

## Abstract

**Highlights:**

**What are the main findings?**

**What is the implication of the main finding?**

**Abstract:**

Background: Little is known regarding the non-inferiority of new anti-seizure medications (ASMs) in terms of replacing valproic acid (VPA) in patients with idiopathic generalized epilepsy (IGE). We hypothesized that replacement of VPA with new ASMs would offer non-inferior or better control of seizure frequency. The purpose of this study was to compare epileptic seizure frequency between the subset of patients with IGE who were on VPA and the subset of patients with IGE who replaced VPA with new ASMs. Methods: Patients with IGE who were on or had been on VPA between January 2016 and March 2022 were divided into a group that replaced VPA with new ASMs (VPA-replace group) and a group that remained on VPA (VPA-continue group). We then compared the groups in terms of seizure frequency and myoclonus. Results: Of the 606 patients on VPA between January 2016 and March 2022, 156 patients with IGE were enrolled to this study (VPA-replace group, *n* = 68; VPA-continue group, *n* = 88). The VPA-replace group included significantly more females than the VPA-continue group (*p* < 0.001). The VPA-replace group also showed significantly higher seizure frequency before replacement (*p* < 0.001), but not after replacement (*p* = 0.074). Patients on monotherapy displayed improved seizure frequency with new ASMs (*p* < 0.001). Among the new ASMs, perampanel (PER) significantly improved seizure frequency (*p* = 0.002). Forty-two patients in the VPA-replace group who had myoclonus achieved significant improvements (*p* < 0.001). Among these, patients on PER monotherapy (*p* < 0.001) or PER + lamotrigine (0.016) showed significantly improved myoclonus scale scores. Conclusions: This study shows the non-inferiority of new ASMs compared to VPA, with better seizure control using new ASMs in subsets of patients with IGE on monotherapy.

## 1. Introduction

Anti-seizure medications (ASMs) have continued to develop, and third-generation ASMs are now available [1]. The second- and third-generation ASMs are collectively termed “new ASMs” [2,3]. These new ASMs are considered to incur fewer adverse effects (AEs) and offer higher tolerability [4], along with advantages including fewer drug–drug interactions and fewer negative impacts on cognitive function [5,6]. Old ASMs are still in use around the world [7,8], and the non-inferiority of new ASMs has been shown for some older ASMs [9].

Even though physicians are generally aware of the advantages of the new ASMs, some old ASMs are still recognized as efficacious. Among the old ASMs, valproic acid (VPA) is one of the representative mainstay ASMs in the real world [10,11,12,13]. However, considering the advantageous and disadvantageous characteristics of each ASM, since ideal ASMs should offer sufficient efficacy with minimal AEs, physicians naturally wonder whether they should continue to prescribe VPA or instead replace VPA with new ASMs.

VPA is frequently used to treat idiopathic generalized epilepsy (IGE) [14,15,16], for which it is considered a first-line agent [17]. Even though strong evidence remains lacking regarding the adequate use of ASMs for IGE [18], many large case series have reported the efficacy of VPA for IGE [16,19]. It is not unusual for patients with IGE to be on VPA.

Thus, since physicians are aware of the efficacy of VPA for IGE, VPA continuation is considered as one treatment option. On the other hand, since physicians are also aware of its possible AEs, replacement of VPA with new ASMs is considered as another treatment option.

The efficacy of new ASMs for IGE has been reported in some studies [20,21,22,23], but little is known regarding the non-inferiority of new ASMs in terms of replacing VPA in patients with IGE. We therefore hypothesized that the replacement of VPA with new ASMs would offer non-inferior or better control of seizure frequency.

The purpose of this study was to compare epileptic seizure frequency between the subset of patients with IGE on VPA and the subset of patients with IGE who underwent replacement of VPA with new ASMs.

## 2. Methods

### 2.1. Study Design and Ethics Approval

The ethics committee at Seirei Hamamatsu General Hospital approved the protocol for this study (approval no. 3959), which was performed in accordance with the principles of the Declaration of Helsinki.

This was a retrospective observational study comparing two groups from a single tertiary center. We performed adjustment of ASMs for the purposes of treatment, but not for the purposes of performing the study.

The design and analysis of this study were conducted in accordance with the Strengthening the Reporting of Observational Studies in Epidemiology (STROBE) statement checklists [24].

### 2.2. Settings

We retrospectively reviewed our electronic medical records and identified patients who were on or had experienced VPA therapy between January 2016 and March 2022 at Seirei Hamamatsu General Hospital, Shizuoka, Japan.

### 2.3. Participants

We included patients diagnosed with IGE, including juvenile myoclonic epilepsy, juvenile absence epilepsy, and exhibiting generalized tonic–clonic seizures. Among these, we divided the patients into two groups: a group comprising patients who discontinued VPA and started new ASMs (VPA-replace group), and a group comprising patients who continued VPA without any administration of new ASMs (VPA-continue group). The inclusion criteria for the VPA-replace group were as follows: (1) patients who underwent replacement of VPA with new ASMs between June 2016 and December 2017; (2) follow-up for ≥2 years after replacement. The inclusion criteria for the VPA-continue group were as follows: (1) patients who had been on VPA throughout the follow-up period from January 2016 to March 2022; (2) no administration of ASMs during follow-up. June 2016 was used for the VPA-replace group because most new ASMs became available in Japan from the end of May 2016. December 2017 was used for the VPA-replace group, because some new ASMs need longer titration periods [25]. Patients who were on VPA after December 2017 and met the criteria were enrolled into the VPA-continue group. Since severe AEs might appear within the first 3 months of starting ASM replacement, and the longest interval between hospital visits in this study was every 3 months, some patients needed more than a year to complete the change from VPA to new ASMs. We therefore set a 1.5-year replacement period from June 2016 to December 2017.

In our facility, in terms of the protocol for VPA replacement, (1) all patients who were on VPA received an explanation from pharmacists about the potential AEs of VPA; (2) female patients ≥14 years old or who had reached menarche received an explanation from pharmacists about potential teratogenicity, gynecological disorders, etc., and were recommended to replace VPA with a new ASM by physicians with respect to patient preference; (3) new ASMs (i.e., lamotrigine (LTG), levetiracetam (LEV), topiramate (TPM), and perampanel (PER)) were chosen based on the guidelines of the Japanese Society of Neurology, in addition to the currently available lacosamide (LCM). In the process of replacing VPA with new ASMs, VPA was tapered off, and new ASM monotherapy was encouraged in our protocol. Regarding the choice of new ASM, patients received descriptions of the advantages and disadvantages from the physician, and the ASM was prescribed based on the preferences of the patient. Ultimately, the new ASM was selected depending on the constitution of the patient, such as the presence or absence of allergies, depression, dizziness, and current medications.

### 2.4. Data Sources

#### 2.4.1. Primary Outcome Measurement

We compared the VPA-replace and VPA-continue groups in terms of seizure frequency. Seizure frequency was evaluated at the latest visit in the period between January 2020 and March 2022.

#### 2.4.2. Secondary Outcome Measurements

We reviewed which new ASMs were used instead of VPA. We also compared seizure frequency between pre- and post-replacement with the new ASMs in the VPA-replace group. For patients with myoclonus and absence seizures, reductions in myoclonus and absence seizures were evaluated.

### 2.5. Variables

#### 2.5.1. Definition of Seizure Frequency

Seizure frequency in this study was graded as follows: daily = 5; weekly = 4; monthly = 3; yearly = 2; seizure-free for >1 year but ≤2 years = 1; seizure-free for >2 years = 0 [26]. Seizures in this study consisted of generalized tonic–clonic convulsions.

#### 2.5.2. Myoclonus Score

We used Ikeda’s myoclonus score [27,28], consisting of five levels from 0 to 4: marked myoclonus, causing incapacity = 4; severe myoclonus, with clear disturbance of daily activities = 3; moderate myoclonus, with some disturbance of daily activities = 2; mild myoclonus without disturbance of daily activities = 1; and absence of myoclonus = 0.

#### 2.5.3. Absence Seizures

We diagnosed a patient as having absence seizures when a patient with IGE clearly showed clinical absence seizures and electrographic generalized epileptiform discharges on electroencephalography with video monitoring. We did not count any symptoms such as subjective feelings of a momentary loss of consciousness in the absence of eyewitnesses.

### 2.6. Bias

When enrolling participants, we used our electronic database. For data collection, hospital staff not involved in this study collected data from the participants. The keywords “patients who are or were on VPA” and “from January 2016 to March 2022” were used to select the participants, and then their age as of March 2022 (end of the follow-up period), sex, hospital identification number, and date of birth were provided to the authors.

For the diagnosis of epilepsy, physicians who were both certified as epileptologists by the Japan Epilepsy Society and had electroencephalogram certification from the Japanese Society of Clinical Neurophysiology (A.F., K.S., H.E.) diagnosed the epilepsy for each patient.

### 2.7. Statistical Analysis

Values of *p* < 0.05 were considered to indicate significant differences in all analyses. The Mann–Whitney U-test was used to compare each group of nonparametric pairs, but not the paired *t*-test, as testing showed non-normal distributions. All statistical analyses were performed using SigmaPlot version 14.5 software (Systat Software, San Jose, CA, USA).

## 3. Results

### 3.1. Participants

Of the 606 patients on VPA during the period between January 2016 and March 2022, 449 patients were excluded. Exclusions were due to non-IGE conditions such as epileptic encephalopathy and neuralgia in 371 patients, incomplete ASM replacement during the enrollment period in 37, and loss to follow-up in 41. As a result, 157 patients with IGE were enrolled in this study.

### 3.2. Clinical Information and Descriptive Data

Among the 157 patients with IGE, the VPA-replace group included 68 patients. The VPA-continue group included 88 patients, excluding one patient who used VPA for concomitant psychiatric disorder. Overall, 156 patients (68 VPA-replace, 88 VPA-continue) met the inclusion criteria (Figure 1).

The VPA-replace group included 42 females and 26 males, ranging in age from 11 to 59 years (mean, 25.8 years; median, 22 years; standard deviation, 11.3 years). The VPA-continue group included 18 females and 70 males, ranging in age from 12 to 72 years (mean, 28.1 years; median, 24 years; standard deviation, 12.2 years) (Table 1). These two groups showed a significance in sex (*p* < 0.001), but not in age (*p* = 0.177).

### 3.3. Outcome Measurements

#### 3.3.1. Primary Outcomes

Comparison of seizure frequency before replacement showed significantly worse control in the VPA-replace group than in the VPA-continue group (*p* < 0.001). However, the same comparison after replacement was not significant (*p* = 0.074). Comparing before and after ASM replacement in the VPA-replace group, replacement with new ASMs improved seizure frequency (*p* < 0.001) (Table 2).

#### 3.3.2. Secondary Outcomes

The 68 patients of the VPA-replace group included 47 patients on VPA monotherapy and 21 patients on polytherapy. In terms of seizure frequency, no significant difference was seen between the mono- and polytherapy subgroups (*p* = 0.161).

All 47 patients on VPA monotherapy replaced VPA with new ASM monotherapy. Among these, 20 patients started PER, 14 started LEV, 9 started LTG, 3 started TPM, and 1 started LCM. Patients with monotherapy showed improved seizure frequency with the new ASMs (*p* < 0.001). Among the new ASMs, PER significantly improved seizure frequency (*p* = 0.002) (Table 3). Polytherapy was not statistically analyzed, due to the complicated process required and the insufficient numbers of patients.

Among the 156 enrolled patients who satisfied the criteria, 47 patients (30%) used to experience or currently experienced myoclonus. Among those 47 patients, 41 patients were assigned to the VPA-replace group, and their myoclonus improved significantly with new ASMs (*p* < 0.001). Among these, 14 patients started PER, 7 started LEV, 4 started LTG, 1 started TPM, 1 started LCM, 8 started combination PER/LTG, 2 started PER/LEV, 1 started LEV/TPM, 1 started LTG/CLB, 1 started LEV/TPM/LCM, and 1 started LTG/PER/TPM. Patients on PER monotherapy (*p* < 0.001) and on PER/LTG (0.016) showed significant improvements in myoclonus scale score (Table 4).

Among the 156 enrolled patients who satisfied the criteria, 11 patients (7%) had experienced absence seizures. Among those 11 patients, 5 patients were in the VPA-replace group; these 5 patients did not show deterioration of absence seizures after changing to new ASMs (1 female on LEV and LTG, 1 female on LEV, 1 female and 1 male on PER, and 1 male on LTG). The six patients in the VPA-continue group comprised one female and five males.

## 4. Discussion

### 4.1. Key Results

The results showed (1) non-inferiority between the VPA-replace and VPA-continue groups in terms of seizure control, even though seizure control was poorer in the VPA-replace group than in the VPA-continue group before replacement; and (2) better seizure control with new ASMs in subsets of the monotherapy group. These results support our hypothesis that replacement of VPA with new ASMs offered non-inferior or better control of seizure frequency in this retrospective single-center case–control study.

### 4.2. Interpretation

The VPA-replace group comprised patients with poorer seizure control and a higher proportion of female patients than the VPA-continue group. This suggests that physicians may have regarded VPA as providing fair efficacy, but tried to achieve better seizure control with new ASMs or tried to avoid AEs from VPA in female patients, due to the known teratogenicity and associations with AEs such as cognitive disorders, autism spectrum disorder in children born while on VPA, ovarian issues, weight gain, and hair loss [29,30,31,32,33,34]. We cannot clarify why the VPA-replace group exhibited poorer seizure control than the VPA-continue group. IGE is a relatively easy disease to control with ASM compared to other epilepsy syndromes, and up to 85% of patients with juvenile myoclonic epilepsy can achieve seizure control on ASMs [35], The International League Against Epilepsy (ILAE) has also reported that IGE responds well to treatment using an appropriate ASM [36]. IGE therefore tends to be considered a benign syndrome [35]. However, the remaining 15% of cases are drug-resistant, and vagus nerve stimulation is also an option [37]. In our study, such cases of drug-resistant IGE might have been enrolled into the VPA-replace group.

Replacement of VPA with new ASMs appears reasonable given these results, even though this study was retrospective in design and the evidence level was therefore relatively low. Replacement of VPA with new ASMs might be encouraged, because the seizure control achieved with new ASMs was generally non-inferior to VPA, and even better than VPA in some subsets of monotherapy groups—especially with PER.

In terms of myoclonus, new ASMs offered better control than VPA. However, we must pay attention to the exacerbation of myoclonus by new ASMs [38]. Considering the efficacy of PER for myoclonic epilepsy in not only IGE [20,21,22], but also progressive myoclonus epilepsy [39,40], along with our results of better control of both seizures and myoclonus in this study, PER might be a good choice among new ASMs for replacing VPA. According to Franceschetti et al. [41], the mechanism by which PER reduces myoclonus is considered to be a reduction in local synchronization and better control of distant synaptic effects.

As PER was the most used ASM in this study, and superiority among the new ASMs was not within the scope of the present research, we cannot form any conclusions as to the actual efficacy of PER for IGE. However, PER may have potential as an appropriate new ASM for patients with IGE, and further investigation is warranted.

The number of patients with absence seizure was small, so we only show the results for reference. We need a greater number of patients in order to evaluate the efficacy of new ASMs in terms of absence seizures.

The pathological mechanisms underlying IGE are only partially understood, and might be related to gamma-aminobutyric acid (GABA)ergic neurons [42]. A relationship between cortical hyperexcitability and VPA has been reported [43,44]. However, some new ASMs have mechanisms of action that are also related to GABA receptors, α-amino-3-hydroxy-5-methyl-4-isoxazole-propionic acid (AMPA) [45], and other receptors [46,47], which may explain the non-inferiority of their effects compared to those of VPA. However, the underlying mechanisms cannot be explained by this study. The fact that sex differed between the VPA-continue and VPA-replace groups was probably due to the VPA replacement protocol in our facility. We therefore considered that the pathological mechanisms might not be related to sex differences.

### 4.3. Limitations and Generalizability

One of the limitations in this study was that this was a retrospective single-center case–control study. Thus, the level of evidence was not high. To obtain a higher level of evidence, a randomized controlled study is required. To perform a double-blinded, randomized, controlled trial, we could randomly divide patients with IGE on VPA into VPA-continue and VPA-replace groups with each new ASM. However, patients with IGE in whom seizures are well controlled on VPA—particularly males without the need to consider teratogenicity, etc.—adults of both sexes who are on VPA and are socially independent might not be candidates for such a clinical trial, due to the risk of seizure recurrence. Thus, potential future studies could be biased due to factors such as age, sex, and social status.

Another limitation of this study was that since our protocol encouraged the replacement of VPA with a new ASM, our policy avoided high doses and long-term use of VPA. This bias should be considered in this study.

Given the low level of evidence, the ability to generalize the present results is quite limited. However, the present findings are relevant when considering replacing VPA with new ASMs in patients with IGE.

## 5. Conclusions

This study showed the non-inferiority of new ASMs compared to VPA in terms of seizure control, even though seizure control was poorer in the VPA-replace group than in the VPA-continue group before replacement. Furthermore, better seizure control was achieved using new ASMs in the PER monotherapy subgroup among patients with IGE.

## Figures and Tables

**Figure 1 jcm-11-04582-f001:**
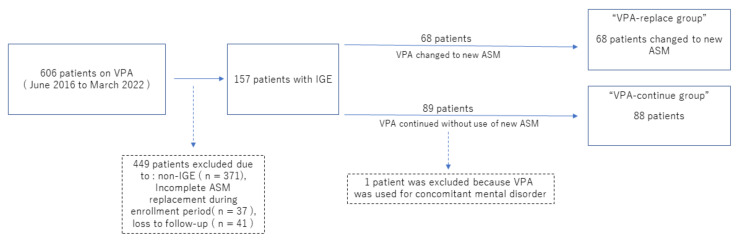
Participants: From January 2016 to March 2022, 606 patients used valproic acid (VPA). Among these, 156 patients diagnosed with idiopathic generalized epilepsy (IGE) were included in this study, and were divided into two groups: One group comprised patients in whom VPA was replaced with a new ASM (VPA-replace group). The other group comprised patients who continued on VPA without replacement with new ASMs (VPA-continue group).

**Table 1 jcm-11-04582-t001:** Clinical information.

	VPA-Replace Group	VPA-Continue Group	*p*-Value
Number of patients	68 (43.6%)	88 (56.4%)	n/a
Sex (Female:Male)	42:26	18:70	<0.001 *
Age (mean, range, SD)	28.1, 12–72, 12.2	25.8, 11–59, 11.3	0.17

n/a: not available; SD: standard deviation; VPA; valproic acid; Statistical signicicance *.

**Table 2 jcm-11-04582-t002:** Seizure frequency in the VPA-replace and VPA-continue groups.

	before Replacement (Mean, Range, SD, 95%CI)	after Replacement (Mean, Range, SD, 95%CI)	*p*-Value
VPA-replace group	0.87, 0–4, 1.11, 0.49–1.13	0.21, 0–2, 0.48, 0.05–0.26	<0.001*
VPA-continue group	0.13, 0–4, 0.52, 0.00–0.27		n/a
*p*-value	<0.001 *	0.074	

95%CI:95% Confidence Interval; SD: standard deviation; VPA: valproic acid; SD: standard deviation; *: signicicant corelations to seizurefrequency; Mann-Whitney Rank Sum Test; n/a: not available.

**Table 3 jcm-11-04582-t003:** Seizure frequency with each new ASM in the VPA-replace group for monotherapy.

AMS and No. of Patients	before Replacement	after Replacement	*p*-Value
Overall 47 patients (mean, range, SD)	0.77, 0–4, 1.11	0.19, 0–2, 0.45	<0.001 *
PER 20 (43%) [mean, range, SD]	1.2, 0–4, 1.4	0.2, 0–2, 0.52	0.002 *
LEV14 (30%) [mean, range, SD]	0.5, 0–3, 0.9	0.1,0–1,0.27	0.067
LTG9 (19%) [mean, range, SD]	0.33, 0–1, 0.5	0.3, 0–1, 0.5	1.000
TPM3 (6%) [mean, range, SD]	0.33, 0–1, 0.56	0.33, 0–1, 0.56	na
LCM1 (2%) [mean, range, SD]	na	na	na

PER: perampanel; LEV: levetiracetam; LTG: lamotrigine; TPM: topiramate; LCM: lacosamide; na: not available; SD: standard deviation; *: significant correlations to the seizure frequency; Mann-Whitney Rank Sum Test.

**Table 4 jcm-11-04582-t004:** Myoclonus scale score after replacement in the VPA-replace group.

AMS and No. of Patients	before Replacement	after Replacement	*p*-Value
Overall 41 patients (mean, range, SD)	1.095, 0–3, 0.43	0.333, 0–2, 0.526	<0.001 *
PER 14 (33% )[mean, range, SD]	1.29, 0–3, 0.61	0, 0–0, 0	<0.001 *
LEV 7 (17%) [mean, range, SD]	1, 1–1, 0	0.714, 0–1, 0.488	0.500
LTG 4 (10%) [mean, range, SD]	0.75, 0–1, 0.9	0.5,0–1,0.577	1.000
TPM 1 (2%) [mean, range, SD]	na	na	na
LCM 1 (2%) [mean, range, SD]	na	na	na
PER/LTG8 (20%) [mean, range, SD]	1, 1–1, 0	0.125, 0–1, 0.354	0.016 *
PER/LEV 2 (4%) [mean, range, SD]	1, 1–1, 0	0, 0–0, 0	na
LEV/TPM 1 (2%) [scale score]	1	1	na
LTG/CLB 1 (2%) [scale score]	1	1	na
LEV/TPM/LCM 1 (2%) [scale score]	1	1	na
LTG/PER/TPM 1 (2%) [scale score]	1	0	na

PER: perampanel; LEV: levetiracetam; LTG: lamotrigine; TPM: topiramate; LCM: lacosamide; CLB: clobazam; na: not available; SD: standard deviation; *: significant correlations to the seizure frequency; Mann-Whitney Rank Sum Test.

## Data Availability

The data supporting the findings of this study are available from the first author (A.F.) upon reasonable request.

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
