# Peer review of "Replacement of Valproic Acid with New Anti-Seizure Medications in Idiopathic Generalized Epilepsy"

_jcm, 2022, doi:10.3390/jcm11154582_

Round 1

Reviewer 1 Report

This work employed a retrospective, single-center cace-control design, to investigate the effects of new AEMs to replace VPA. It has somewhat scientific interests, however, still remains a few of major questions.  Significance of the work was insufficient, which should be enhanced to indicate the contributions of the work to the scientific mechanism / clinical value of IGE. Gender effecs would be considered and discussed carefully.

Author Response

Responses to Comments from the Reviewers

Thank you very much for reviewing our manuscript. We greatly appreciate the advice provided.

Reviewer #1

1- how long had the patients been on valproate before the replacement?

Response: Thank you very much for raising this point. We were also interested in and tried to collect this information. However, since almost all patients had been referred from other facilities, some referral letters lacked the required data. We therefore could not obtain precise information about the period of VPA use for all patients.

2- What is the selection criteria that you chose to shift patients from valproate to other ASM?

Response: Thank you very much for raising this point, which we had overlooked. We have stated the protocol for the replacement of VPA with new ASMs in the revised text (Section 2.3 Participants).

3- What about absence seizures, did you replace those patients to other ASM as levetiracetam for example considering that it is not indicated in this type of epilepsy?

Response: We only had 11 patients who clearly showed absence seizures with clinical and electrographic evidence. Six of these 11 patients were still on VPA. This might be because VPA is efficacious for absence seizure, as you pointed out. However, the other 5 of the 11 patients were in the VPA-replacement group and did not show deterioration of absence seizures on the new ASMs, including levetiracetam. We have added this data to the revised manuscript. However, as the volume of patients was small, we have also added some explanation to the Discussion section.

Reviewer 2 Report

Dear authors

1- how long had the patients been on valproate before the replacement? 

2- What is the selection criteria that you chose to shift patients from valproate to other ASM?

3- What about absence seizures, did you replace those patients to other ASM as levetiracetam for example considering that it is not indicated in this type of epilepsy?

Author Response

Responses to Comments from the Reviewers

Thank you very much for reviewing our manuscript. We greatly appreciate the advice provided.

Reviewer #2

Thank you for inviting me to review this manuscript. The paper itself is well written, although somewhat descriptive. In this retrospective study, The authors have conducted the non-inferiority of new ASMs compared to VPA in terms of seizure control, even though seizure control was poorer in the VPA-replace group than in the VPA continue group before replacement, and better seizure control using new ASMs in subsets of patients with IGE on monotherapy. However, the manuscript need some modification.

1-In the method please clarify, What was the selection criteria for replacing VPA with new ASMs? was it based on clinical features such as the duration of the disease, seziure frequency, history of ASMs, ...

Response: Thank you very much for pointing out this omission. We have added the protocol for changing from VPA to new ASMs in the protocol of the revised text (Section 2.3 Participants).

2-It is recommended that in the discussion section explain more about why the VPA-replace group comprised patients with poorer seizure control.

Response: Thank you very much for the opportunity to discuss poorer seizure control. Some patients with IGE prove intractable. We have discussed this point with some citations.

Reviewer 3 Report

Thank you for inviting me to review this manuscript.The paper itself is well written, although somewhat descriptive. In this retrospective study,The authors have conducted the non-inferiority of new ASMs compared to VPA in terms of seizure control, even though seizure control was poorer in the VPA-replace group than in the VPA continue group before replacement,and  better seizure control using new ASMs in subsets of patients with IGE on monotherapy. However, the manuscript need some modification.

In the method please clarify, What was the selection criteria for replacing VPA with new ASMs? was it based on clinical features such as the duration of the disease, seziure frequency, history of ASMs, ...

It is recommended that in the discussion section explain more about why the VPA-replace group comprised patients with poorer seizure control. 

Author Response

Thank you very much for reviewing our manuscript. We greatly appreciate the advice provided.

Reviewer #3

1-Unless I have misunderstood the meaning of parts of the text, you seem to have first collected over 600 instances of patients treated with valproate between 2016 and 2022, and then discarded roughly ¾ of them, mainly because none had idiopathic generalised epilepsy. Yet all the cases of valproate treated idiopathic generalised epilepsy that you studied were apparently recruited in 2015 and 2016, with none occurring subsequently, which is most surprising as I do not think the disorder is ceasing to occur elsewhere.

Response: We are sorry for any confusion resulting from our lack of explanation. We have therefore added that “Patients who were on VPA after December 2017 and met the criteria were enrolled into the VPA-continue group” (Section 2.3 Participants).

2-In your table 2 I cannot work out the comparisons to which your two cited P values apply. There seems to be 3 possible comparisons, and more if you supplied values for seizure frequency at the outset in the valproate-unchanged group. Are you comparing seizure occurrence rates before and roughly 5 years after valproate withdrawal in the same patients, between seizure occurrence rates before valproate withdrawal and occurrence rates some five years later in those continuing to take the drug, or between rates five years after withdrawal and after non-withdrawal in ä different set of patients. And even if you clarify this and can justify why events that have not been analysed between the starting and finishing times may not have confounded the interpretations, your argument for non-inferiority seems to demonstrate non-inferiority of the new drugs only at the P < 0.05 level, but inferiority of the new drugs at the 10% level of confidence, which is not very persuasive.

There is no information regarding valproate dosage changes in those continuing to take the drug over several years, and this may have complicated the interpretability of your data.

Response: Thank you very much for raising these very important points. We apologize for not fully showing the statistical comparisons, as pointed out. We have added comparisons and confidence intervals between pre- and post-seizure frequency for the VPA-replace group, which showed a significant difference (p < 0.001).

Reviewer 4 Report

Unless I have misunderstood the meaning of parts of the text, you seem to have first collected over 600 instances of patients treated with valproate between 2016 and 2022, and then discarded roughly ¾ of them, mainly because none had idiopathic generalised epilepsy. Yet all the cases of valproate treated idiopathic generalised epilepsy that you studied were apparently recruited in 2015 and 2016, with none occurring subsequently, which is most surprising as I do not think the disorder is ceasing to occur elsewhere.

In your table 2 I cannot work out the comparisons to which your two cited P values apply. There seems to be 3 possible comparisons, and more if you supplied values for seizure frequency at the outset in the valproate-unchanged group. Are you comparing seizure occurrence rates before and roughly 5 years after valproate withdrawal in the same patients, between seizure occurrence rates before valproate withdrawal and occurrence rates some five years later in those continuing to take the drug, or between rates five years after withdrawal and after non-withdrawal in ä different set of patients. And even if you clarify this and can justify why events that have not been analysed between the starting and finishing times may not have confounded the interpretations, your argument for non-inferiority seems to demonstrate non-inferiority of the new drugs only at the P < 0.05 level, but inferiority of the new drugs at the 10% level of confidence, which is not very persuasive.

There is no information regarding valproate dosage changes in those continuing to take the drug over several years, and this may have complicated the interpretability of your data.

Author Response

Responses to Comments from the Reviewers

Thank you very much for reviewing our manuscript. We greatly appreciate the advice provided.

Reviewer#4

Reviewer 1 : This work employed a retrospective, single-center cace-control design, to investigate the effects of new AEMs to replace VPA. It has somewhat scientific interests, however, still remains a few of major questions. Significance of the work was insufficient, which should be enhanced to indicate the contributions of the work to the scientific mechanism / clinical value of IGE. Gender effecs would be considered and discussed carefully

Response:

Thank you very much for this suggestion. We appreciate this opportunity to discuss the “scientific mechanisms / clinical value of IGE”. Since this retrospective study did not initially reach the level of discussing the scientific mechanisms underlying IGE, we would have added some clinical perspectives regarding IGE in the Discussion section.

Round 2

Reviewer 1 Report

The authors have adequately addressed all concerns raised by this reviewer. I would be pleased to see this publised.

Author Response

Comments from the reviewers
Reviewer#1
The authors have adequately addressed all concerns raised by this reviewer. I would be pleased to see this publised.
Response: Thank you very much for your kind comments. We appreciate your reviewing time and your opinions.

Reviewer 4 Report

Of the 3 matters that I raised previously, I think you have added material that clafrifies my first issue regarding the absence of valproate exposed pregnancies after about 2016 from the valproate-changed group.

I still find Table 2 difficult to interpret - in the left column you seem to be comparing the values in the two rows, and then comparing left column row 2 with the number in the rright column, yet showing P values in the same rightmost column.

You do not seem to have addressed my point about changed valproate dosages (or other ASMs) in the continuing valproate group, yet dosages  of newer ASMs presumably were adjusted in the valproate changed group. This, and the fact that no information is provided about baseline seizure control in the valproate-unchanged group, leave a sense that continuing valproate therapy may not have been managed toachieve optimal results, bet the new ASM therapy managed with that aim. It would be useful if you could reject that possibility.

Author Response

Comments from the reviewers
Revewer#4
Of the 3 matters that I raised previously, I think you have added material that clafrifies my first issue regarding the absence of valproate exposed pregnancies after about 2016 from the valproate-changed group.
Response: We are sorry for not fully replying to your comments. It seems that just one out of the three comments came to me. We sincerely apologize and would like to reply to them.

I still find Table 2 difficult to interpret - in the left column you seem to be comparing the values in the two rows, and then comparing left column row 2 with the number in the rright column, yet showing P values in the same rightmost column.
Response:
We agree with your opinion. The P-value is comparison between the two groups. However, the current Table 2 seems to compare the seizure frequency before and after the replacement. This is our mistake. We would like to replace rows and columns.

You do not seem to have addressed my point about changed valproate dosages (or other ASMs) in the continuing valproate group, yet dosages of newer ASMs presumably were adjusted in the valproate changed group. This, and the fact that no information is provided about baseline seizure control in the valproate-unchanged group, leave a sense that continuing valproate therapy may not have been managed to achieve optimal results, bet the new ASM therapy managed with that aim. It would be useful if you could reject that possibility.
Response:
Thank you very much for raising the point. We should have explained this point more precisely. We would like to state that “In the process of VPA replacement with new ASMs, VPA was tapered or tapered off and new ASM monotherapy was encouraged in our protocol”. In terms of the VPA-continue group, we understand the point that was raised. This bias might be one on the study limitations. We would like to add some comments in Discussion part as one of the study limitations.
